# Approaches to Configuration Determinations of Flexible Marine Natural Products: Advances and Prospects

**DOI:** 10.3390/md20050333

**Published:** 2022-05-19

**Authors:** Zong-Qing Huo, Feng Zhu, Xing-Wang Zhang, Xiao Zhang, Hong-Bao Liang, Jing-Chun Yao, Zhong Liu, Gui-Min Zhang, Qing-Qiang Yao, Guo-Fei Qin

**Affiliations:** 1State Key Laboratory of Generic Manufacture Technology of Chinese Traditional Medicine, Lunan Pharmaceutical Group Co., Ltd., Linyi 273400, China; huozq163@163.com (Z.-Q.H.); zhufeng0716@163.com (F.Z.); lianghongbao1985@163.com (H.-B.L.); yaojingchun@lunan.cn (J.-C.Y.); clevertree@163.com (Z.L.); lunanzhangguimin@163.com (G.-M.Z.); 2State Key Laboratory of Microbial Technology, Shandong University, Qingdao 266237, China; 3College of Pharmacy, Shandong University of Traditional Chinese Medicine, Jinan 250355, China; szyzhangxiao@163.com; 4Institute of Materia Medica, Shandong First Medical University & Shandong Academy of Medical Sciences, Jinan 250000, China; 18810337655@163.com

**Keywords:** flexible marine natural products, configuration determinations, X-ray single-crystal diffraction, NMR-based methods, circular dichroism-based methods, quantum computational chemistry-based methods, chemical transformation-based methods

## Abstract

Flexible marine natural products (MNPs), such as eribulin and bryostatin, play an important role in the development of modern marine drugs. However, due to the multiple chiral centers and geometrical uncertainty of flexible systems, configuration determinations of flexible MNPs face great challenges, which, in turn, have led to obstacles in druggability research. To resolve this issue, the comprehensive use of multiple methods is necessary. Additionally, configuration assignment methods, such as X-ray single-crystal diffraction (crystalline derivatives, crystallization chaperones, and crystalline sponges), NMR-based methods (JBCA and Mosher’s method), circular dichroism-based methods (ECCD and ICD), quantum computational chemistry-based methods (NMR calculations, ECD calculations, and VCD calculations), and chemical transformation-based methods should be summarized. This paper reviews the basic principles, characteristics, and applicability of the methods mentioned above as well as application examples to broaden the research and applications of these methods and to provide a reference for the configuration determinations of flexible MNPs.

## 1. Introduction

Oceans are a treasure trove for the discovery of novel small molecules and drugs. To date, more than 35,000 marine natural products (MNPs) have been discovered, and this number is growing at a rate of about 1500 per year (more than 400 papers) [1]. According to the statistics, 71% of the skeletons of new compounds can only be found in marine resources [2]. Meanwhile, it is worth mentioning that there are 13 MNP-sourced drugs that are entering the EU and US drug markets, 4 of which have been approved in the past 3 years [3]. In addition, four drugs of marine origin or containing marine ingredients are in phase III clinical trials; eight are in phase II clinical trials; and twenty are in phase I clinical trials [3,4]. Among them, a considerable number include molecules with flexible backbones, such as eribulin, bryostatin, taltobulin, plocabulin, and monomethyl auristatin E (Figure 1) [5]. Flexible MNPs are attracting considerable critical attention in the development of modern marine drugs.

It is well known that during druggability research on natural products, correct structures are crucial for determining the structure–activity relationship, structural modifications, and total synthesis. However, due to the multiple chiral centers and geometrical uncertainty of flexible systems, it is still challenging to determine the configuration determinations of flexible MNPs. Over the past few decades, numerous strategies have been steadily developed to solve this problem. Those methods include X-ray single-crystal diffraction, NMR-based methods, circular dichroism-based methods, quantum computational chemistry-based methods, and chemical transformation-based methods [6,7,8,9]. In this review, the basic principles, characteristics, and applicability of the abovementioned methods as well as application examples will be presented to provide a reference for the configuration determinations of flexible MNPs.

## 2. X-ray Single-Crystal Diffraction (XRSCD)

X-ray single-crystal diffraction (XRSCD) is regarded as the most reliable strategy for absolute configuration determination. Differences in the X-ray anomalous scattering effect of each atom are used to determine the absolute configuration of molecules and can provide the precise spatial position of all of the atoms in a compound in the solid state, including how the atoms are connected, the molecular conformation, and accurate bond length and bond angle data [10]. The strength of the anomalous scattering effect is proportional to the electron cloud density of the atom, which is manifested as the stronger anomalous scattering effect of the atom with a larger atomic number [11,12]. The anomalous scattering effect is also positively related to the wavelength of the irradiated radiation. Cu-kα or Mo-kα radiation is usually selected as the light source in single-crystal diffraction experiments. Due to the longer wavelength of Cu-kα rays, its scattering ability is 6–10 times that of Mo-kα. If the molecule to be tested contains heavy atoms, both Cu-kα and Mo-kα radiation can be used to determine its absolute configuration [12]. However, most of the molecules in natural products are mainly composed of C, H, O, and N and do not contain heavy atoms, so their absolute configurations can only be assigned when Cu-kα is used as a radiation source. Furthermore, for crystalline compounds that are devoid of strong scatterers, the combined X-ray/ECD (solid-state CD/TDDFT approach) technique is another effective tactic that can be implemented to solve absolute configuration issues [13]. In crystal structure analysis, the Flack parameter *x* (0 < *x* < 1) and its standard uncertainty *u* are introduced to determine the absolute configuration of the crystal. If *u* is small (*u* < 0.04) and the Flack parameter *x* is close to zero (generally *x* < 0.1), it indicates that the configuration is correct; otherwise, the correct configuration is its enantiomer [11]. Nevertheless, the bottleneck of this technique is the requirement of preparing a single crystal that is of sufficient size, quality, and stability. Unfortunately, for flexible molecules, it is difficult to generate a suitable crystal because of their special properties (structure, solubility, and molecular weight). Accordingly, several approaches based on X-ray single-crystal diffraction have emerged to address this limitation.

Bastimolide A (**1**), a macrolide isolated from marine cyanobacterium *Okeania hirsute* in 2015, possesses a flexible skeleton [14]. Since bastimolide A shows viscous properties in various solvents, it is impossible to crystallize. Thus, derivatization strategies were smoothly pursued by Shao and colleagues to generate crystalline compounds from an organic solvent [14]. Four derivatives (**1a**–**1d**) were prepared, three of which were acetonide derivatives, and one of which was a nona-*p*-nitrobenzoate derivative (Figure 2). The latter (**1d**) generated suitable crystals, and X-ray single-crystal diffraction was carried out to determine its absolute configuration: 9*S*, 11*R*, 15*S*, 19*R*, 21*R*, 23*R*, 27*S*, 31*R*, 35*R*, 39*S* (Figure 2). This approach of derivatizing the original molecule to improve its crystallization properties is well established, and Burns and co-workers successfully assigned the relative and absolute structures of alkenes by means of common Mo X-ray analysis using crystalline osmate derivatives [15].

Further, crystallization chaperones can also be used to induce the co-crystallization of target molecules. Crystallization chaperones are molecules that bind covalently or non-covalently to the analyte, helping it to assemble into a crystalline lattice that is then generally applied for protein analysis [16,17]. The Richert group discovered that tetraaryladamantanes (TAAs) are easily co-crystallized with small molecules in the form of inclusive complexes [18]. A total of 3 TAAs were selected as crystallization chaperones for 52 small molecules to evaluate their ability to induce crystallization (Figure 3). A total of 88% of the small molecules were crystallized, and 77% of them obtained high-resolution crystal structures [18]. The results showed that this crystallization chaperone had a great crystallization ability and good versatility. Although marine natural products were not contained in the test sets, several representative flexible compounds, such as **2b**–**2f** and **2s**–**2w**, have demonstrated the potential and suitability of this method to resolve the absolute configuration of flexible skeletons [18].

In addition, crystalline sponges (CS), single-crystalline porous coordination networks that do not involve sample crystallization, have been proposed. They contain large and regular cavities that can absorb small molecules in the same way that sponges can and arrange them in an orderly manner so that their structures can be determined with X-ray diffraction [19,20,21].

Elatenyne (**3**) is a marine natural product that possesses a special pseudo mirror-symmetric skeleton and whose absolute configuration cannot be determined by regular methods because of its pseudo-meso core structure [22]. Crystalline sponges greatly contributed to Fujita et al.’s absolute-structure determination of **3** [23]. Additionally, to confirm the accuracy of this method in determining absolute configurations, a crystalline sponge analysis of a click adduct (**3a**) was performed, achieving positive results. It is worth mentioning that the total amount required for this experiment was only 100 µg, and 95 µg could be easily recovered after the experiment, showing the obvious advantages of using this method.

Using this method, the absolute configuration of miyakosyne A (**4**), a cytotoxic and flexible acetylene isolated from a marine sponge *Petrosia* sp., was revealed (Figure 4) [19,24]. However, when comparing the synthetic stereoisomers and subsequent chemical degradation of **4**, it was found that the absolute configuration at C14 should be *R* instead of *S* [19,25,26] (Figure 5). Although there was an error in the judgment of the absolute configuration of **4**, the accuracy of the other compounds that were tested was appreciable. Therefore, by continuously improving CS, we believe that this method will be groundbreaking in determining the absolute configurations of complex natural products.

## 3. NMR-Based Methods

### 3.1. J-Based Configuration Analysis (JBCA)

*J*-based configurational analysis (JBCA) was first reported by Murata and Yasumoto in 1995 and was further formalized by Murata in 1999 and has now become a reliable and efficient tactic that is an extensively used method for determining the relative configuration of 1,2 and 1,3-methine systems, whose chiral carbons can be substituted by C, O, N, Cl, S, or other groups [27,28,29,30,31,32]. In this method, a set of specific coupling constants (^3^*J*_H,H_, ^3^*J*_H,C_ and ^2^*J*_H,C_) that has been divided into three ranges (large (L), medium (M), and small (S), according to their magnitude, Table 1) provides key geometric information (such as dihedral angles) for conformation assignment. Thus, the relative configuration can be determined through assigning the conformation by matching the experimentally measured coupling constants with a priori estimations of all of the possible rotamers.

A general workflow diagram of the configuration assignment in a 1,2-methine system is presented in Figure 6. First, the values of ^3^*J*_H,H_ and ^2,3^*J*_H,C_ should be measured. The former can be easily extracted with a ^1^H-^1^H NMR experiment, whereas the latter is only able to undergo complex data analysis after heteronuclear correlation experiments, such as 2D hetero half-filtered TOCSY (HETLOC), phase-sensitive HMBC (PS-HMBC), or HSQC-TOCSY [33,34,35,36]. If the value of ^3^*J*_H,H_ is judged as S or M, then the *threo* or *erythro* configuration can be assigned through fitting the measured values to the a priori estimated magnitude of the coupling constants A-1, A-2, B-1, B-2, A-2/A-3, A-3/A-1, B-2/B-3, or B-3/B-1 in Figure 7. Nevertheless, the same magnitude of coupling constants between A-1/A-2 and B-1/B-2, and the other two exceptions (one is the case of all three staggered conformers coexisting with comparable populations, and the other is the condition of a given rotational conformer deviates more than 15° from a staggered one [29], which are not mentioned in Figure 7) will restrict the application of JBCA. Fortunately, these three situations rarely reside in natural products with multiple substituted acyclic structures [27]. When ^3^*J*_H,H_ is shown as typical L, NOE/ROE experiments are indispensable due to the coupling constants having the same magnitude. As long as the proton of C1 relates to the proton of C4, it means that C1 and C4 are in *gauche* alignment, indicating that the main conformer is A-3 and that it is B-3 otherwise.

In a 1,3-methine system, the entire skeleton is split into two moieties (C2-C3 and C3-C4), and the proton on the methylene (C3) in a higher field is designated as H3, while the other is denoted as H3′. Among the two moieties, there are twelve single dominant conformers (Figure 8) and twelve equilibrating rotamers (Figure 9). Following the flow diagram summarized in Figure 10, all of the conformers belonging to the two moieties can be unequivocally identified and further attributed in the configuration as *syn* or *anti*.

Ostreol B (**5**), a cytotoxic and polyhydroxy compound with a long and flexible carbon skeleton, was isolated from cells that were cultivated from the toxic dinoflagellate *Ostreopsis* cf. *ovata* collected in South Korea [37]. The configurations of C13–C22 were successfully assigned using the JBCA method. The configuration of C13–C14 was assigned as *threo* on the basis of ^3^*J*_H13, H14_ = 2.9 Hz, ^2^*J*_H13, C14_ = −1.1 Hz, ^3^*J*_H13, C14_ = 2.0 Hz, and ^3^*J*_H14, CH3_ = 6.2 Hz. As shown in Figure 11, the experimentally obtained coupling data indicated that C15/C18 and H17b/16−OH had an *anti* orientation, while the coupling constants ^3^*J*_H17a, H18_ = 3.0 Hz, ^3^*J*_H17b, H18_ = 9.2 Hz, ^2^*J*_H17a, C18_ = −1.1 Hz, ^2^*J*_H17b, and C18_ = −5.6 Hz were sufficient to assign an *anti* relationship between C16/C19 and H-17a/18-OH. Thus, the configuration of C16–C18 was assigned as *erythro*. C18–C19 was set to a *threo* configuration on the basis of the four small coupling constants of ^3^*J*_H18, H19_, ^2^*J*_H18, C19_, ^2^*J*_H19, C18_, and ^3^*J*_H18, C20_. For the C19–C21 fragment, the JBCA of C19/C20 and the coupling constants of ^3^*J*_H20a, H21_, ^3^*J*_H20b, H21_, ^2^*J*_H20a, C21_, and ^2^*J*_H20b, C21_ allowed the *anti* orientation to be assigned to C19/C22 and H20b/21−OH, thereby suggesting a *threo* configuration between C19 and C21. Thus, the absolute configurations of C13–C22 were assigned as 13*R*, 14*R*, 16*R*, 18*S*, 19*S*, and 21*S* based on the stereochemistry of the 14*R* and 16*R* centers, which were assigned through chemical transformation and Mosher’s method. In addition, the configurations within C24–C28 were determined by ROE observation. Moreover, the configuration of C28–C43 was also determined by JBCA, and their main conformations were assigned as shown in Figure 11.

### 3.2. Mosher’s Method

Mosher’s method was first proposed by Professor Mosher in 1973 and has now become a mature and reliable absolute configuration determination method for secondary alcohols due to the improvements that have been made to it over the past few decades [38,39]. Firstly, secondary alcohols react with (*R*)- and (*S*)-*α*-methoxy-*α*-(trifluoromethyl)phenylacetic acid (MTPA) to generate (*R*)- and (*S*)-MTPA esters, respectively [40]. As shown in Figure 12A, the trifluoromethyl group, the carbonyl group of MTPA, and the *α*-H of secondary alcohol are located in the same plane (Mosher plane, the blue color). Additionally, due to the shielding effect of benzene, the proton chemical shifts of the L2 moiety in (*R*)-MPTA ester are smaller than those in the (*S*)-MTPA ester, so the Δ*δ*_SR_ (*δ*_(*S*)-MTPA ester_-δ_(*R*)-MTPA ester_) values of L2 should be positive, and the Δ*δ*_SR_ values of the L1 moiety should be negative (in the Mosher ester, determining the absolute configuration of the chiral carbon using the Δ*δ* of *β*H alone is known as the classical Mosher’s method) [41]. Comprehensively considering the Δ*δ* of the protons of the L1 and L2 moieties is more reliable and is the most commonly used modified Mosher’s method [42]). Then, by calculating the Δ*δ* values of the Mosher esters, the moiety with “−” Δ*δ* values should be placed on the left side of the Mosher plane, and the part with “+” Δ*δ* values should be placed on the right side of the plane (Figure 12B). Finally, IUPAC nomenclature rules are applied to determine the absolute configuration of the secondary alcohols. Shao and colleagues provide a comprehensive standard operating guide for this method [43].

When using the modified Mosher’s method, the positive and negative Δ*δ* values of the Mosher derivatives of the secondary alcohols must be regularly arranged on both sides of the chiral center, and the secondary hydroxyl group with large steric hindrance must be converted to the opposite side [44]. As many Mosher’s agents have been developed [45,46,47,48,49], special attention should be paid to the Δ*δ* (Δ*δ*_SR_ or Δ*δ*_RS_?) in practical applications. For example, when the -OH groups of (*S*)-MTPA were replaced with –Cl, the MTPA-Cl with an *R* absolute configuration was generated (Cahn–Ingold–Prelog (CIP) sequence: -OCH_3_ > -CF_3_ > -COOH > -Ph, -OCH_3_ > -COCl > -CF_3_ > -Ph); thus, the Δ*δ* of the MTPA-Cl ester should be Δ*δ*_RS_ (*δ*_(*R*)-MTPA-Cl ester_-*δ*_(*S*)-MTPA-Cl ester_) (Figure 12A). In addition, researchers need to be very cautious when using MTPA as a derivatizing reagent because of the very small differences in the chemical shift that are caused by its conformation issues. This means that in some cases, and especially when applied to secondary alcohols, the confidence in the results obtained by the MTPA-based Mosher’s method is not sufficient. In contrast, methoxyphenylacetic acid (MPA), one of the most commonly used ester-forming reagents, has a larger chemical shift difference than MTPA, which leads to it having higher confidence in determining the absolute configuration. Therefore, MPA is clearly superior to MTPA as a derivatizing agent for secondary alcohols [50,51].

Trichophycin C (**6**), a highly functionalized polyketide possessing a characteristic chlorovinylidene moiety, was isolated from *Trichodesmium* bloom material collected from the Gulf of Mexico [52]. The absolute configurations at C4 and C10 in its structure were determined using the modified Mosher’s method. The authors used MTPA as the ester-forming reagent to generate the bis-MTPA esters of **6** and calculated the Δ*δ*_SR_ values. As shown in Figure 13, the Δ*δ*_SR_ values are regularly arranged on both sides of the chiral center. Additional “−” Δ*δ* values for H-1, H-2, H-3, H-11, H-13a, H-15, H-19, and H-20 together with “+” Δ*δ* values for H-4, H-5, H-6a, H-6b, H-7, H-8, H-9, H-21, and H-22 revealed absolute configurations of 4*S* and 10*R*.

## 4. Circular Dichroism (CD)-Based Methods

### 4.1. Exciton Chirality CD (ECCD)

Exciton chirality CD (ECCD) is a non-empirical and reliable theorizing method that relies on the direct association of the absolute configuration and ECD through the exciton coupling of chromophores (ECD reporter groups) [53,54,55]. It is suitable for chiral systems with two or more non-conjugated chromophores going through electric dipole-permitted transitions [56]. Exciton coupling is initiated by the staggered electric transition dipole moments (TDMs) of two chromophores, which is shown as two Cotton effects with opposite signs in the ECD spectrum. When the positive Cotton effect is localized in the long wavelength range, it indicates that the two TDMs are clockwise from a forward perspective, which is defined as positive chirality (Figure 14). Thus, the relationship of the ECD and stereo-structure information in the molecules is established.

The ECCD analysis workflow follows the following procedures [56]: The first step is to measure the ECD spectra of the molecule with suitable chromophores and to determine the type of exciton coupling (positive chirality or negative chirality). The next step is to determine the dominant conformers of the candidate configurations. Then, the TDM orientation in each chromophore should be assigned accurately. Once the above steps are all completed, the assignment of the configuration can be carried out. As shown in Figure 14, the two TDMs of the chromophores in the dominant conformation of the candidate configuration were placed in front of each other, meaning that their rotation direction (clockwise or counterclockwise) can then be obtained by establishing a sense of rotation to make the front TDM agree with the back one. Thus, the absolute configuration can be assigned by comparing the rotation directions of the TDMs vs. the exciton chirality of the ECD.

ECCD is not as simple as it seems, and it requires careful analysis and extra attention in many respects. Choosing suitable chromophores is the first crucial point to consider before deciding to use ECCD. An eligible chromophore must be capable of electronic π–π* transition. This can be inherent in the molecule or introduced through a chemical reaction. Additionally, the method can still be used when the two chromophores are different. Commonly used chromophoric systems include bis(benzoates), biaryls, benzene derivatives, tetraarylporphyrins (TPP), polyenones, enones, allylic benzoates, and other moieties that are associated with the π–π* component [56]. Benzoic acid-related moieties and TPP are often introduced as chromophores to assist in the determination of molecular configurations. In particular, TPP is suitable for the polyoxygenated polycyclic skeletons of marine natural products that are commonly isolated from dinoflagellates because of its ultra-long-range (40–50 Å) coupling properties [57].

Moreover, the TDM direction within each chromophore should be determined accurately. It is closely related to the symmetry of the chromophore structure. The TDM directions of commonly used chromophores have been studied intensively [58]. However, for some inherent chromophores, the TDM direction is still vague, and their TDM directions should be corroborated by quantum mechanical computations. Besides this, the conformations of candidate molecular configurations also need to be carefully analyzed, and if necessary, a systematic conformational search can be performed to obtain complete conformational information [56].

In addition to what has been mentioned above, the composition of the bonds in the ECD spectrum should also be clarified. The spectrum involves three main aspects: (a) Is the π–π* transition a major contributor to ECD? (b) Is the wavelength region in which exciton coupling occurs truncated by the cutoff wavelength of the solvent? (c) Does electric–magnetic exciton coupling (π–π* transition coupled with n–π* transition) occur?

The absolute configuration assignment of phycotoxin gymnocin B (**7**) is a classic ECCD application case. Gymnocin B was originally isolated from the dinoflagellate *Karenia mikimotoi* and possesses a long polyoxygenated polycyclic skeleton with five flexible seven-membered rings [59]. Since the weak reactivity between its secondary hydroxyl groups and MTPA-Cl, the use of Mosher’s method is restricted [60]. Considering the ability of long-range coupling and the low detection threshold of TPP, researchers chose it as the ECD reporter group to settle the charity issue of C10 and C37 (Figure 15). TPP chromophores can be linked to weak reactive hydroxyl groups with relatively high yields using a combined acryloylation/cross-metathesis approach. Additionally, a comprehensive conformational analysis of the simplified TPP derivatives (**7b**) was performed using molecular mechanic calculations with an MMFF94s force field and Monte Carlo conformational search. Three lowest-energy conformations were obtained, with the Boltzman-weighted populations consisting of 86%, 9%, and 5%, respectively. The interporphyrin twists for the first two conformations were judged as positive, which is consistent with the positive exciton charity that is shown as a positive split Cotton effect (λ (MeOH) 419 nm (Δε +11), 414 nm (Δε −15)) that was observed in the ECD spectra (Figure 15B,C). Thus, the absolute configurations of C10 and C37 were determined to be *S* and *S*, respectively.

### 4.2. Complexation-Induced CD (ICD)

#### 4.2.1. Snatzke’s Method (Mo_2_(OAc)_4_-Induced CD)

Snatzke’s method, first proposed by Snatzke in 1981, is a convenient and effective empirical approach by which to determine the absolute configuration of cyclic or acyclic 1,2-diols (*prim/sec*, *prim/tert*, *sec/sec*, *sec/tert*, *tert/tert*) [61,62]. When mixing *vic*-diol with a solution comprising an achiral reagent Mo_2_(OAc)_4_ (Figure 16A) and during the subsequent measurement of the ICD spectra of the chiral complexes, if the ligating structure had a negative O-C-C-O dihedral (anticlockwise), the Cotton effect at around 310 nm was negative; instead, when it had a positive O-C-C-O torsional angle (clockwise), the Cotton effect at around 310 nm was positive (the helicity rule) [62]. Thus, the absolute configuration of the *vic*-diols can be determined by the sign of CD at around 310 nm of its Mo_2_-complex.

Although this method was first proposed to solve the configurational assignment of rigid *vic*-diols, Górecki’s group and Frelek’s group further developed it to include flexible vicinal diols [62,63]. First, it is necessary to make sure that the relative configuration of the *vic*-diols is *threo* or *erythro* through JBCA, NOE, or a reaction with acetone. If the relative configuration of the *vic*-diols is *threo*, then the favorable conformation in the Mo_2_-complexes of the *vic*-diols is the one in which both of the O-C-C-R units have an antiperiplanar orientation (Figure 16B). Nonetheless, in the *erythro*-1,2-diols complexes, there are two possible arrangements for the diol unit, which could lead to the decisive CD having opposite signs for the same absolute configuration (Figure 16C). In this situation, the favorable conformation is that the -R group, which has an O-C-C-R unit with an antiperiplanar orientation, has the biggest steric hindrance. Additionally, if it is difficult to judge which group has the biggest steric hindrance, it is better to refer to the ICD of similar compounds with a known absolute configuration [63].

It should be noted that other functional groups in the 1,2-diols, such as esters, amides, and ethers, do not interfere with the results. However, in the presence of the carboxylic acid group, several hydroxy and/or amino groups would prevent its employment [64]. On the other hand, if there are chromophores in the 1,2-diols, their inherent contributions should be subtracted from the ICD spectra of the chiral complexes.

Bacillcoumacins B (**8**) and C (**9**), two amicoumacin-type isocoumarin derivatives, were isolated from the marine-derived bacterium *Bacillus* sp. [65]. Their side chain contains *vic*-diol structural fragments. In order to determine the relative configuration of the H-8′ and H-9′ of compound **8**, it was conversed to acetonide **8a**. The NOE correlations between H-8′ and H-9′ and from both H-8′ and H-9′ to the methyl protons of acetonide depicted an *erythro*-diol. The *erythro* configuration of the diol in **9** was assigned by the same *J*_H-8′/H-9′_ value of **8** and by the literature. Then, their absolute configurations were determined to be 8′*s* and 9′*S* by Mo_2_(OAc)_4_-induced CD (Figure 17), respectively. On the basis of the negative Cotton effects at 300 nm, their Mo_2_-complexes favored the b2 conformation in Figure 16C.

#### 4.2.2. Rh_2_(OCOCF_3_)_4_-Induced CD

Rh_2_(OCOCF_3_)_4_-induced CD is especially suitable for the absolute configuration determination of secondary and tertiary alcohols. The CD spectra of the Rh_2_(OCOCF_3_)_4_ complexes of chiral alcohols have five Cotton effects (A: 590–585 nm, B: ~500 nm, C: 460–455 nm, D: ~420 nm, and E: ~350 nm) ranging from 270 to 600 nm. The E band can be used to determine the absolute configuration of the secondary or tertiary alcohols by applying the empirical bulkiness rule [66,67]. As shown in Figure 18, when the sign of the Cotton effect for the E band is negative, the structure of the secondary alcohol is of the ‘*bR*’ type, and when it is positive, the structure of the secondary alcohol is of the ‘*bS*’ type (according to the size of the substituents, L represents a large substituent and M represents a medium substituent). Thus, the position assignment of each group can be preliminarily determined, and then the absolute configuration of the chiral alcohols can be determined by the CIP system.

If there are multiple functional groups in the alcohols, and especially if the additional functional group is in the vicinity of the chiral hydroxy group, then this method should be applied with caution [67,68]. In this case, the site of complexation and the privileged complexation should be analyzed.

Thalassosamide (**10**), a new cyclic trihydroxamate compound, is a siderophore discovered from the marine-derived bacterium *Thalassospira profundimaris* [69]. The 2*R*, 2′*R*, and 2″*R* absolute configuration was established by means of Marfey’s method using fluorodinitrophenyl-5-_L_-leucine amide (FDLA) as the bifunctional reagent. To elucidate the absolute configuration of C9, C9′, and C9″, rational chemical degradation was performed to obtain three identical diesters, **10a,** with original configurations. Then, the absolute configurations of C9 in **10a** was deduced as *R* by the negative E band at 350 nm in the Rh_2_(OCOCF_3_)_4_-induced CD spectrum. Thus, the stereo configuration in the original compound is 9*R*, 9′*R,* and 9″*R* (Figure 19).

Recently, chirality sensing using stereodynamic probes has been successfully applied to determine the absolute configuration of natural products [70,71]. It relies on the strong and characteristic chiroptical readout that is induced by the molecular interactions between a nonracemic chiral substrate and a chromophoric, CD-silent probe. A covalent or non-covalent binding event that conforms to well-defined asymmetry-induced processes can efficiently transfer chiral substrate information (chiral guest) to the stereodynamic sensors (achiral host), thereby leading to a strong Cotton effect in the ultraviolet region of the latter [72]. The molecular configuration can be determined by matching the sign of the Condon effect with the twist direction of a specific structure in the probe. Commonly used probes include bridged biaryls, imine foldamers, trityl propellers, stereodynamic metal complexes, and porphyrins. Among them, the porphyrin-based method is also known as the tweezers method, which can be used to analyze samples with micromolar or sub-micromolar concentrations [9,73]. Additionally, bridged biaryl probes have been applied to natural products, as mentioned at the beginning of the paragraph. Although this method has been proposed and used for many years, to date, there are practically no cases of it being used to determine the absolute configuration of flexible MNPs. However, it performed well when dealing with flexible molecules, and it has the potential to be an interesting approach.

## 5. Quantum Computational Chemistry-Based Methods

With the continuous progress of related theories and technologies, quantum chemical theoretical calculation methods used to assist in structure identification have become standard methods for confirming the structure of complex natural compounds. By calculating the specific optical rotation (SOR), NMR chemical shift, electronic circular dichroism (ECD), or vibrational circular dichroism (VCD) of a chiral molecule in order to compare it to the measured values, its relative and absolute configurations can be determined.

### 5.1. NMR Calculations for Marine Natural Products

Following a series of papers on the calculation of NMR parameters for natural products published in the early 2000s by the Bifulco, Köck, Bagno, and Sebag group, a comprehensive paper covering the state-of-the-art in the quantum chemical calculations of NMR parameters in natural product chemistry was published by Tantillo in 2012 [74,75,76,77,78]. Additionally, in more recent years, with the continuous expansion and deepening of related research, the calculation models for determining NMR parameters have experienced significant improvements in terms of their accuracy, reliability, and scope of application, especially in the configurational assignment of natural products [79]. This can be applied not only to determine the molecular configurations but also to the review and correction of previously reported ones [79]. It must be noted that, in general, computational NMR can only determine the relative configuration unless the absolute configuration of a key position has been determined.

Commonly involved NMR parameters for this method are ^1^H and ^13^C chemical shifts and spin–spin coupling constants (SSCCs). SSCCs are often used to determine the relative configuration of the local area, while the calculation of the chemical shift of ^13^C is more widely applied [78]. Besides assisting in carbon signal attribution, SSCCs can also determine the relative configuration of the whole molecule.

The general workflow of NMR calculation includes conformational searches, geometry optimization, the calculation of NMR properties, molecular energy calculations, and Boltzmann averaging, as well as the comparison of the calculated values with those obtained from experiments. It is crucial to choose an appropriate statistical method to evaluate the calculation results in the final step. The statistical methods include classical tactics (e.g., absolute difference, MAE, CMAE, RMSD, and R^2^) and emerging tactics (e.g., CP3, DP4, DP4+, and *J*-DP4), by which the correct isomer can be assigned [79,80,81]. The DP4 method, proposed by the Goodman group in 2010, has been the most widely used tool in this field because of its ability to handle problems where only one set of experimental data (^1^H or ^13^C chemical shifts) corresponds to multiple candidate structures [82]. It is worth mentioning that there are also two automated versions of this method, PyDP4 and DP4-AI [83,84]. On the basis of this method, Sarotti proposed DP4+ in 2015, which is highly flexible, easy to use, and can handle diverse chemical structures and isomerizations [85]. In addition, *J*-DP4 incorporates coupling constants into the DP4 analysis, with three modes: d*J*-Dp4, I*J*-DP4, and i*J*/d*J*-DP4 [86]. The performance of this strategy is 2.5 times better than the original DP4 method [86].

Due to the complexity and diversity of the conformational isomers of flexible molecules, when performing NMR calculations, the conformational set of each isomer that is randomly generated by the program can be used to replace the calculated conformational distribution, and their DP4/DP4+ probability can then be calculated. This step should be repeated a specified number of times, and the average can be taken to determine the final probability of each isomer [87]. In addition, Hehre and co-workers describe an efficient method for calculating the ^13^C NMR chemical shifts of flexible natural products that has been tested with numerous cases [88].

In the above section discussing Mosher’s method, the determination of the C4 and C10 chiral center configurations of trichophycin C (**6**) was discussed as a typical case [52]. The configurations of its other two chiral centers (C5 and C7) were assigned by comparing experimentally obtained SSCCs with theoretically calculated values. To better explain the flexibility of the molecule, geometry optimization was performed at two different functionals (B3LYP and M062X). As clearly shown in Table 2, the RMSD between the theoretically calculated and experimentally measured *J*-couplings is the lowest for C7*R*C5*R*. The “error bars” on the theoretical values were also estimated through comparing the average coupling constants obtained from the conformational distributions that were calculated at the two different functionals and received positive feedback. These results highlight C7*R*C5*R* as the stereoisomer for **6****,** and good agreement is also shown in Figure 20 when combined with 10*R* and 4*S*.

In addition to calculating the three classical NMR parameters mentioned above, interproton distances, residual dipolar couplings (RDCs), and residual chemical shift anisotropies (RSCAs) can also serve as calculations. In several candidate structures, the computational predictions of the interproton distances can be complementarily applied to the NOE/ROE information, facilitating accurate and reliable configuration assignments. RDCs and RSCAs are anisotropic NMR parameters that provide angular data about carbon–hydrogen bonds and shielding tensors, respectively. In order to observe RDCs and RCSAs, the molecules must be partially aligned, usually with the help of an orienting media such as confined polymer gels or liquid crystals. For rigid natural product molecules, these two methods can solve configuration issues based on simple calculated models. However, for flexible molecules, simple models are often ineffective. Fortunately, this has been solved to some extent by using RDCs as constraints. Progressive stereo locking (PSL), an RDC-based force field method proposed in 2017, can effectively solve the relative configuration problems of flexible molecules [89]. Luy and colleagues established a method based on time-averaged molecular dynamics with dipole coupling as a tensor orientation constraint, and this method can be used to solve molecular structure problems of any size [90]. More recently, a method called floating chirality restrained distance geometry (fc-rDG) calculations was used to directly evolve structures from NMR data such as NOE or RDCs [91]. In contrast to RDCs, RCSAs have been rarely used, mainly due to the practical difficulty of extracting RCSA from the spectra, as dissolving the analyte in the orienting media introduces both isotropic and anisotropic chemical shift changes [92,93,94,95,96].

### 5.2. ECD Calculations for Marine Natural Products

In the above section, we discussed some reliable and well-established semi-empirical methods based on ECD spectroscopy used to determine the absolute configuration of chiral molecules. Additionally, in recent years, increasing attention has been drawn to addressing the configuration issues resulting from using TDDFT (time-dependent density functional theory) for ECD computation. It is even possible to state that this method has become a standard tactic in the study of natural products. Having a high intuitive ability and clarity is a hallmark of this method. The absolute configuration can be assigned by directly comparing the computationally predicted ECD spectrum with the experimental one.

The typical process for calculating ECD follows several steps. The first step is to perform a conformational analysis on the tested molecule. The dominant conformation that contributes more to the calculation of the ECD spectrum is selected by performing corresponding calculations with appropriate functionals and basis sets that have been selected according to the molecular weight and properties of the molecule. Following this, TDDFT-ECD calculations can be performed on each dominant conformation to obtain a spectrum, and a Boltzmann-weighted average of all of the obtained spectra is then calculated to generate the predicted ECD spectrum of the assumed target configuration. Finally, the absolute configuration of the molecule can be determined by comparing the generated ECD spectrum with the experimental one [54,97,98,99].

Molecules with flexible skeletons usually possess multiple conformations in solution, and their experimental ECD spectra represent the average of those multiple conformations in time and space. Therefore, it is crucial to conduct rational conformational analysis for this class of molecules. The complex conformational distribution of flexible molecules means a significant increase in computational costs and time. In general, natural products from the ocean feature greater flexibility and a higher molecular weight, leading to a longer computation time or a higher probability of errors in the computations. Consequently, it is necessary to combine other methods for comprehensive analysis when necessary.

Fiscpropionates A (**12**) and B (**13**), two polypropionate derivatives featuring an unusually long hydrophobic chain, were isolated from the deep-sea-derived fungus *Aspergillus fischeri* FS452 [100]. The authors determined their relative configurations using a combination of NOESY and coupling constants. Then, the absolute configurations of **12** and **13** were established as 9*R*, 10*R*, 12*S*, 14*S*, and 16*S* and 9*R*, 10*R*, 12*S*, and 14*S*, 16*S* via quantum chemical calculations of the ECD spectra at the B3LYP/6-311+G(d,p) level (Figure 21).

### 5.3. VCD Calculations for Marine Natural Products

For natural products whose chromophore is far away from the chiral center or does not contain a chromophore, the corresponding Cotton effect on the ECD spectrum is extremely weak or even non-existent, so it is difficult to use semi-empirical ECD methods or ECD calculations to determine the absolute configuration. Since molecular vibrations are universal, all chiral natural products theoretically feature unique VCD spectra, and the VCD spectrum is better able to reveal subtle conformational and stereochemical differences, so VCD-based methods can be used to settle the absolute configuration of chiral molecules. Nevertheless, the effective and direct interpretation of VCD spectra is very difficult due to the complexity of the molecular vibrational modes. At present, the most effective and reliable method is to compare the calculated VCD spectra of different isomers with the measured spectra to determine the absolute configuration of the chiral compounds [54].

While ordinary DFT methods, rather than the more complex TDDFT, can be applied to calculate VCD spectra, the computation cost remains huge for flexible-skeleton molecules [54,101]. Additionally, the higher accuracy will also inevitably lead to an increase in the calculation time. In addition, the acquisition of the VCD experimental spectrum relies on a relatively high sample concentration (20–50 mg/mL) and a long measurement time (several hours) because of the weak intensity of the VCD band [54,102]. Therefore, it is difficult for this method to be widely used in the determination of the absolute configuration of natural products with limited quantities.

Hemicalide (**14**), a mitotic inhibitor with high antiproliferative potency against human cancer cell lines at subnanomolar concentrations, was isolated from the marine sponge *Hemimycale* sp. Collected in the deep water around the Torres Islands (Vanuatu) [103], it features a 46-carbon atom skeleton containing 21 stereocenters. Degradation and derivatization cannot be performed on its trace amount (ca. 1 mg). Relative configurations of C8-C13 and C18-C24 were reasonably assigned by means of comparison with the data obtained from synthesized diastereomeric model compounds and further NMR analysis and calculations [104,105]. In addition, combining stereocontrolled synthesis with NMR, IR, and VCD analyses, the relative configuration of C36-C46 was determined (Figure 22). Specifically, the configuration of C42 was settled by comparing the computational (**14a** and **14b**) and experimental (sample **I** and **II**) spectra of VCD after the relative configurations of other chiral centers had been assigned. As shown in Figure 22, the IR of the two candidate isomers (**14a** and **14b**) was evaluated, and no specific differences were found, while the VCD was sufficiently characteristic at 1070–1170 and 1350–1400 cm^−1^. Good agreement is shown in Figure 22C, and this agreement allows for the unambiguous assignment of **II** to **14b** and **I** to **14a**. According to the NMR match between the prototype molecule of **I** and **14**, the relative configuration of C42 was assigned as shown in the structure of **14a**.

## 6. Chemical Transformation-Based Methods

Chemical transformation, or, more specifically, the synthesis and degradation of the molecule, has always been an indispensable method in elucidating the structures of natural products, especially when identifying the configuration of marine natural products, which often contain structural features such as long aliphatic chains and abundant functional groups. Although this method cannot directly determine the configuration of the molecule, it can provide an essential prerequisite for other strategies.

Needless to say, the most ideal solution would be to synthesize various isomers of the tested molecule to compare spectroscopic data. Of course, researchers may synthesize the corresponding configuration after a few years of painstaking effort, but this intellectually difficult and physically tedious approach is not ideal. From a practical standpoint, for difficult-to-synthesize compounds, the best option is to synthesize decisive segments for comparison. For example, amphidinol 3 (AM3) (**15**) is a marine natural product that is produced by the dinoflagellate *Amphidinium klebsii* [106]. Subsequently, in 1999, the absolute configuration was determined by extensive NMR analysis and degradation [107] (Figure 23). Recently, the absolute configuration of C2 and C51 has been revised. Tohru and co-workers chemically synthesized the C31-C67 part of **15** and ultimately identified the absolute configuration of AM3 as 32*S*, 33*R*, 34*S*, 35*S*, 36*S*, and 38*S* by combining the degradation of the original compound [108].

The other option is the appropriate degradation or derivatization of new natural products, followed by spectroscopic analysis to tackle their absolute configuration. The attractive reaction involves acetylation, catalytic hydrogenation, hydrolysis, Baeyer–Villiger oxidation, ozonolysis, periodate oxidation, hydrogenolysis, and other oxidative or reductive chemical reactions [109]. After chemical transformations, the determination of complex molecular configurations becomes relatively straightforward. In this area, since there are a number of excellent reviews that already exist, we will only discuss how Baeyer−Villiger oxidation, ozonolysis, and periodate oxidation may be useful in the context of the structural analysis of natural substrates.

A classic example is in the work directed toward ascolactone A (**16**), which was originally isolated from the marine-derived fungus *Ascochyta salicorniae*. The absolute configuration at C9 was determined by Baeyer–Villiger oxidation with MCPBA followed by the hydrolysis of ester **16a** to **16b** and **16c** [110]. As a result, the configuration of **16c** was obtained as *R* by chiral-phase GC/MS analysis. Combined with NMR analysis data, the configuration of **16** was determined to be 1*R*, 9*R* (Figure 1A).

Ozonolysis was employed to deal with cytotoxic macrolide amphidinolide J (**17**), which was isolated from the cultured dinoflagellate *Amphidinium*, a symbiont of the Okinawan *Amphiscolops* sp. marine flatworm [111] (Figure 1B). The ozonolysis of **17** followed by NaBH_4_ reduction and acetylation yielded peracetates, corresponding to its C1–C7, C8–C11, and C12–C16 fragments (**17a**–**17c**). The absolute configurations of all six stereocenters were determined by comparing NMR, the MS spectra, and the optical rotation symbols of both the synthesized and fragmentary compounds based on **17**.

Periodate oxidation was applied to the structural analysis of phosphoeleganin (**18**), a novel phosphorylated polyketide from ascidian *Sidnyum elegans*. To establish two corresponding fragments, **18a** and **18b**, the compound was oxidatively cleaved with NaIO_4_ followed by NaBH_4_. Additionally, the fragments were then subsequently derivatized to Mosher esters. Ultimately, as a result of these conversions, the absolute configuration of **18** was confirmed [112] (Figure 1C).

## 7. Conclusions

The diversity and complexity of marine ecological environments have provided a rich substrate for marine species diversity, which is considered to be the cradle of pharmacology due to the plethora of promising biological probes and exciting drug candidates. Even so, among the MNP-derived drugs that have been marketed and that are in clinical stages of development, there are some molecules with flexible systems, and the assignment of absolute configurations frequently unable to be securely determined without the application of other technologies.

In this review, six structural analysis techniques based on spectroscopic methods, X-ray diffraction, quantum chemical calculations, and chemical transformations were discussed. We analyzed how each of these methods can be utilized to resolve the configurations of flexible marine natural products and briefly discussed the corresponding examples. Although modern developments in numerous strategies have been able to identify the structures, for most MNPs, especially flexible MNPs, there is no general solution to the absolute configuration (AC) of a compound. It can be said that the determination of the absolute configuration of most flexible MNPs relies on the combined use and cross-validation of multiple methods. Many methods that have been practiced for a long time, such as XRD, JBCA, the classical CD method, and Mosher’s method, are still the main methods used to determine the configuration, and their accuracy and reliability are good.

This paper also pays attention to some emerging strategies, such as quantum computational chemistry, crystallization partners and crystalline sponges, and electron diffraction. Historically, quantum computational chemistry is not very new, and some strategies such as computational NMR are present in the literature that date back to at least the last century. In a manner of speaking, IGLO and GIAO paved the way for the broad use of computed chemical shifts for structure elucidation in organic molecules [78]. Additionally, this method has recently seen a marked increase in accuracy, affordability, and application, something that can be attributed to the rise of advanced technical development, such as residual dipolar couplings (RDCs) and residual chemical shift anisotropies (RSCAs). In particular, the use of RDCs as constraints provides a new route for the solution of flexible molecular configuration problems.

With the rapid development of artificial intelligence-related technologies in recent years, artificial neural networks (ANNs) have also participated in configuration research. Sarotti introduced a conceptually new method based on pattern recognition analysis (PRA) in 2013 by using ANN and ^13^C NMR data to discover misassigned structures [113]. Such ANN-PRA tools provide positive results for the identification of connection and regiochemical errors. For example, this method further confirmed the previously revised structure of annuionone A. However, it is less efficient for stereochemical isomerization reactions and is more subtle in NMR spectral discrimination. Thus, in a further study, Zanardi and Sarotti extended this method with additional stereochemical information provided by ^1^H spectroscopy and 2D C-H correlation experiments to construct three new ANNs (ANN-TMS_vac_, ANN-MSTD_sol_, and ANN-mix) and then revised and proposed the three-dimensional structures of fortucine and stagonolide G, respectively [114].

Both crystallization partners and crystalline sponges are newly established and are based on known X-ray diffraction. Both eliminate the limitation of high-quality crystals. Although there is no successful case of the use of complex flexible MNPs, this is still the method of choice for determining the absolute configuration because of its advantages, including its requirements of a smaller dosage and its simple operation.

A diffraction theory that is more dynamic than the well-known X-ray diffraction is required, and electron crystallography is more complex than kinematic diffraction. Moreover, a specialized detector was used for the experiment, limiting broad adaptability [115]. Thus far, electron crystallography has only played a minor role in structural chemistry. Fortunately, the feasibility of this technology has been confirmed. Electron diffraction has no requirements for the size for the crystal, and it enables the precise structure of each component in the mixture with non-crystalline carbon nanotubes to be determined [116]. Encouraged by these benefits, and although this method is not successful when applied to flexible marine natural products, electron crystallography has the potential to remarkably accelerate and impact the fields of natural product chemistry, drug discovery, and many other fields.

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
