# Peer review of "Approaches to Configuration Determinations of Flexible Marine Natural Products: Advances and Prospects"

_marinedrugs, 2022, doi:10.3390/md20050333_

Round 1

Reviewer 1 Report

The authors present a summary of methods to assign configuration to challenging molecules where motion/flexibility cause complications. Overall, the review is fairly comprehensive and does a great job highlighting the most useful methods. 

There are two areas that could use more detail: 1. MTPA section; and 2. RDC/RCSA.

For the MTPA section, it should also include MPA. Riguera and colleagues originally reported the benefits of MPA in J. Org. Chem. 1996, 61, 8569-8577. Subsequent papers also highlighted that MTPA can be problematic, and that MPA leads to bigger chemical shift differences and higher confidence in the assigned absolute configuration. There have been clear recommendations to use MPA over MTPA to avoid incorrect assignments.

The other section that needs some work is RDC/RCSA section. The authors claim that these methods require harsh conditions, but that is not true. The methods are not harsh, but some conditions make it challenging to retrieve the compound from the alignment media, but those challenges can be overcome. The issue, of course, is that using RDC's and RCSA's has mostly be done by comparing calculated values to experimentally measured including the Science paper cited. So, for flexible molecules neither method works particularly well if modeling is required. However, this has been addressed, at least in part, through the use of RDC's as constraints. One of the early methods was called Progressive Stereo Locking (PSL), ACS Chem. Biol. 2017, 12, 2157-2163. Next, Luy and colleagues published a paper in Chem Sci. 2019, 10, 8774-8791 again supporting the use of molecular dynamics to account for motion. A more recent paper also used floating chirality, which was the part of the PSL approach: Mar. Drugs 2021, 20 (1), 14. At the very least, it should be noted that these newer approaches recognize that addressing motion will help overcome barriers. For molecules that are rigid, RDC and RCSA approaches work fine with simple calculated models. However, rigid structures rarely present challenges for configurational assignment. Given the nature of this review, these newer developments seem appropriate. 

The paper would also benefit from another round of edits for english grammar.

Overall, this review highlights relevant examples of natural products that present challenges due to their flexible nature. These tend to be the structures that are most difficult. So, this is an important area, and this review will be highly relevant.

Reviewer 2 Report

The authors have described well about the various approaches towards the  configuration determination of flexible MNP. This review and its content would definitely please fellow scientists, hence I would like to recommend this for publication.

Reviewer 3 Report

The review by Guo-Fei Qin and co-workers provides a comprehensive and balanced overview of the field. There are a few more methods or approaches which I suggest mentioning. Moreover, since the field of stereochemical analysis of natural products is well-established, many relevant reviews have appeared in recent years, which I recommend quoting as a recognition of colleagues’ work.

Section 1: there are several reviews on the (absolute) structure elucidation of natural products by multiple means which the authors might quote, for instance: Nat. Prod. Rep., 2019, 36, 1005-1030; Curr. Med. Chem. 2018, 25, 287-320.

Section 2: it should be mentioned that crystalline compounds devoid of strong scatterers may still be employed for AC assignment by hyphenated X-ray/ECD technique (Chirality, 2009, 21, S181–S201). The method has been applied to several flexible compounds, see for instance incensol derivatives by Hussain’s group.

Section 3: please consider quoting the extensive review by Riccio and co-workers (Chem. Rev., 2007, 107, 3744-3779).

Section 3.2: other methods related to Mosher’s, but with increased accuracy, should be mentioned; see Chem. Rev. 2004, 104, 17-117.

Section 4.2: there are several methods of “ICD” type other then the mentioned Mo- and Rh-based ones. These are commonly referred to as Cottonogenic derivatives. One example is the tweezers method by Berova and co-workers (reviewed in Chem. Commun. 2009). Another example is the use of dynamic probes reviewed by C. Wolf. Yet another example is biphenyl probes by Superchi’s group (reviewed in Curr. Med. Chem. cited before; most recent example J. Nat. Prod. 2020, 83, 4, 1061–1068).

Section 5.1: it should be clarified that NMR methods provide, in general, only the relative configuration and not the absolute one, unless at least one stereogenic element is independently assigned. Here, Hehre’s protocol (J. Nat. Prod. 2019, 82, 2299−2306) needs to be mentioned.

Minor point: please define ANN-PRA and correct the first acronym (line 640).

Reviewer 4 Report

This review article explains principal methods for determining absolute configuration of flexible marine natural products. The methods include X-ray single crystal diffractions, NMR-based and CD-based techniques, and computational approaches. The article gives a compact and practical guide for applications, from which readers can pick up perspectives of the field and may further read reference for concrete applications. Therefore, the reviewer would recommend publication in the journal, if the authors update the following minor points.
(1) In line 225 of page 9, "mosher" should be "Mosher" since this word stands for the developer's name.
(2) In line 225-226 of page 9, "determination" should be "determining", shouldn't it?
(3) In line 243 of page 10, numbers in atomic groups should be written as subscripts.
(4) In line 249 of page 10, compound 11 is referred to in sentences, but the structure doesn't appear in the paper. Please correct this situation.
(5) In line 257 of page 10, "exciton" should be capitalized.
(6) In line 281 of page 11, "coincide" has a nuance of "happening by chance". The reviewer suggests this word be replaced with "agree".
(7) In line 282 of page 11, "TMDs" should be "TDMs".
(8) In line 285 of page 11, the reviewer believes that "eligibility" may be better with "eligible" in this context.
(9) In line 288 of page 11, the reviewer believes that "chromophores systems" will be better with "chromophoric systems" in general.
(10) In line 310 of page 12, "mian" should be "main".
(11) In line 329 of page 12, "complexation" should be capitalized.
(12) In line 346 of page 13, the reviewer feels weirdness in the words "acetone reaction". Can this be "reaction with acetone"?
(13) In line 373 of page 13, "6" of "d6" should be written as subscript.
(14) In lines 375-392 of page 14, the reviewer could not understand the explanation. The correlation between A-F and 270-600 nm is difficult to follow, which confuses the explanation about E-band. CIP system is also new to readers. Without understanding those points, it is really hard to read sentences. 
(15) In lines 452, 458, and 463 of page 15-16, the authors chose the word "level" to describe the difference in computational methods. The difference in calculation in this context, however, is the choice of functional. Usually, the difference in calculation level is attributable to the depth in basis sets, but the authors use the triple-zeta 6-31g(d,p) for both cases. Could this be corrected?
(16) In Figure 20 of page 16, the functional "M062X" is abbreviated as "M06", which is strange. It should be precise.
(17) In line 489 of page 17, "the compound molecule" should be simply "the molecule".
(18) In lines 491, 494, and 510 of page 17, "spectrums" should be "spectrum".
(19) In lines 547 of page 18, "According the match NMR" should be "According to the NMR match".
(20) In line 556 of page 18, "molecular" should be "molecule".
(21) In line 564 of page 19, the reviewer doesn't think "unimaginable" is a good choice of word here. Doesn't it mean "not ideal"?
(22) In line 577 of page 19, this sentence should be described in the present tense, not in the past tense.
(23) In lines 579-580 of page 19, the reviewer couldn't understand what the authors want to say. It requires more precise and logical explanation.
(24) In line 582 of page 19, shouldn't "used" be "useful" in this sentence?
(25) In line 598-599 of page 19-20, is "relative fragments" correct? Instead, can "corresponding fragments" work in here, perhaps?
(26) In line 600 of page 20, "with mosher ester" should be "to Mosher ester".
(27) In line 616 of page 20, "configuration determinations" can be simply "configurations". What we solve is configuration, not determination.
(28) In line 622 of page 20, "Fortunately" can be removed.
(29) In line 626 of page 20, "Of course" can be removed.
(30) In line 640 of page 21, the appearance of "ANN-PAR" is too sudden for readers to follow. It is more true especially because there is no reference in here. The authors should introduce the full name and the reference. Also, "ANN-PAR" should be "ANN-PRA", shouldn't it?
(31) In line 789 of page 23, a space is missing in "Circulardichroism".
(32) In line 793 of page 23, numbers in "Mo2(OAc)4" should be written as subscript.
(33) In line 827 of page 24, "2010" should be in bold.
(34) In line 861 of page 24, "2016" should be in bold. "52" should be in Italic.
(35) In line 877 of page 25, "J. Org. Chem." should be in Italic.
